# Metabolomics Approach Reveals Important Glioblastoma Plasma Biomarkers for Tumor Biology

**DOI:** 10.3390/ijms24108813

**Published:** 2023-05-16

**Authors:** Adriana C. Ferrasi, Ricardo Puttini, Aline F. Galvani, Pedro T. Hamamoto Filho, Jeany Delafiori, Victoria D. Argente, Arthur N. de Oliveira, Flávia L. Dias-Audibert, Rodrigo R. Catharino, Octavio C. Silva, Marco A. Zanini, Gabriel A. Kurokawa, Estela O. Lima

**Affiliations:** 1Laboratory of Molecular Analysis and Neuro-Oncology, Department of Internal Medicine, Botucatu Medical School, São Paulo State University, Botucatu 18.618-687, Brazil; adriana.ferrasi@unesp.br (A.C.F.); ricardo.puttini@kroton.com.br (R.P.); alinefgalvani@gmail.com (A.F.G.); victoria.argente@unesp.br (V.D.A.); octavio.castro@unesp.br (O.C.S.); gabrielkurokawa@gmail.com (G.A.K.); 2Department of Neurology, Psychology and Psychiatry, Botucatu Medical School, São Paulo State University, Botucatu 18.618-687, Brazil; pedro.hamamoto@unesp.br (P.T.H.F.); marco.a.zanini@unesp.br (M.A.Z.); 3Innovare Biomarkers Laboratory, School of Pharmaceutical Sciences, University of Campinas, Campinas 13.083-877, Brazil; jeanydelafiori@gmail.com (J.D.); arthurnoin95@gmail.com (A.N.d.O.); flaviald.nutricao@gmail.com (F.L.D.-A.); rrc@fcm.unicamp.br (R.R.C.)

**Keywords:** glioblastoma, metabolomics, biomarkers, plasma samples, 5-hydroxymethyluracil, NAPE

## Abstract

Glioblastoma (GB) is the most aggressive and frequent primary malignant tumor of the central nervous system and is associated with poor overall survival even after treatment. To better understand tumor biochemical alterations and broaden the potential targets of GB, this study aimed to evaluate differential plasma biomarkers between GB patients and healthy individuals using metabolomics analysis. Plasma samples from both groups were analyzed via untargeted metabolomics using direct injection with an electrospray ionization source and an LTQ mass spectrometer. GB biomarkers were selected via Partial Least Squares Discriminant and Fold-Change analyses and were identified using tandem mass spectrometry with in silico fragmentation, consultation of metabolomics databases, and a literature search. Seven GB biomarkers were identified, some of which were unprecedented biomarkers for GB, including arginylproline (*m*/*z* 294), 5-hydroxymethyluracil (*m*/*z* 143), and N-acylphosphatidylethanolamine (*m*/*z* 982). Notably, four other metabolites were identified. The roles of all seven metabolites in epigenetic modulation, energy metabolism, protein catabolism or folding processes, and signaling pathways that activate cell proliferation and invasion were elucidated. Overall, the findings of this study highlight new molecular targets to guide future investigations on GB. These molecular targets can also be further evaluated to derive their potential as biomedical analytical tools for peripheral blood samples.

## 1. Introduction

Glioblastoma (GB) is the most frequent primary malignant tumor of the central nervous system (CNS), and only 7.2% of patients survive for five years post-diagnosis [1]. For the initial diagnosis, imaging techniques, such as computed tomography (CT) and magnetic resonance imaging (MRI) with intravenous contrast, are essential guides for planning surgical and radiotherapeutic strategies and the treatment follow-up [2]. Despite advances in strategies for the diagnosis and treatment of CNS tumors, GB management remains challenging. If neuroimaging is not practicable for extensive screening when a clear clinical indication does not exist, patients with brain tumors may experience diagnostic delays, especially in the early stages, in which the clinical symptoms are not well defined. The standard care strategy for GB is surgical resection [3]; however, surgical intervention is limited by the high rate of tumor infiltration, which impairs the total removal of tumor cells. Adjuvant treatment with chemotherapeutic agents, such as temozolomide, often results in relapse, even after a good initial response [4].

Genetic and epigenetic alterations [4], such as IDH1/IDH2 mutations [5], MGMT methylation [6], and 1p19q co-deletion, are relevant for estimating prognoses and appropriate treatments [7]. However, the molecular pathways involved in this disease remain unclear. Further, limiting studies to nucleic acids restricts the possibility of better therapeutic and diagnostic approaches. Thus, investigating molecules that enhance the current understanding of GB biology and shed light on early tumor biomarkers and new signaling pathways for tumorigenesis comprehension, therapeutic intervention, or evaluation of treatment progression is of particular importance [8].

Considering the intratumoral heterogeneity and alterations induced by cancer at the whole-body level, analyses performed on blood samples may reflect the global phenotype of GB. Studies based on the metabolomics of biological matrices, such as plasma, serum, and urine, have revealed differential metabolite profiles in several types of cancer [9]. Metabolites reflect the end point of cellular biochemical interactions as they are by-products of biological processes, representing the possibility of a connection between molecular changes and phenotypes. This linkage highlights the extensive understanding of cellular biochemical pathways and their corresponding outcomes, such as cancer [10].

In this context, the search for metabolites in patients with GB is important, especially for investigations at the whole-body level, which might reveal widespread metabolic changes and highlight possible organic responses to the disease. Thus, the primary aim of this study was to evaluate the differential profile of metabolites in the blood plasma of patients with GB compared with healthy individuals. As our results enabled the identification of GB biomarkers in blood samples, the secondary goals were to clarify the metabolic alterations in these patients and propose biochemical pathways altered in GB biology.

## 2. Results

### 2.1. Epidemiological Characteristics

The control group comprised healthy individuals between 18 and 80 years old (median age of 41.5 years old (range 22–80)) with no comorbidities. This group of 50 participants consisted of 50% female and 50% male volunteers. The GB group comprised 15 male and 9 female patients with GB, with a median age of 57.5 years old (range, 20–79 years). The incidence of GB in males was 1.6 times higher than that in females, in accordance with the incidence rate by sex reported by CBTRUS [11].

### 2.2. Biomarker Selection, Identification, and Distribution

Partial least squares discriminant analysis (PLS-DA) of the GB and CT datasets revealed an evident separation between the groups (Figure 1), which indicates that the set of *m*/*z* (mass-to-charge ratio) features selected by variable importance in projection (VIP) scores could differentiate both groups through the importance of each *m*/*z* feature. Considering a VIP score > 2.5 (Appendix A), the five most important *m*/*z* features for the GB group could be selected: *m*/*z* = 111, *m*/*z* = 143, *m*/*z* 294, *m*/*z* = 819, and *m*/*z* = 931 (Table 1). In addition to VIP scores, *m*/*z* features were ranked using fold change (FC) univariate analysis, with a threshold of FC ≥ 2, represented by Log_2_(FC) > 1.0 (Figure 2). As a complementary result, FC analysis was associated with a Student’s *t*-test (*p*-value < 0.05) represented by a volcano plot graphic (Figure 3). These analyses were performed to observe the most intense *m*/*z* features of the GB group compared with the CT group, which were: *m*/*z* = 931, *m*/*z* = 294, *m*/*z* = 112 (isotope of *m*/*z* = 111), *m*/*z* = 936, and *m*/*z* = 982 (Table 1). Both statistical analyses resulted in a total of seven GB biomarkers. To evaluate the distribution of each metabolite selected by both analyses among individuals, a heatmap was built that revealed a clear difference in biomarker intensity between samples from GB and CT participants (Figure 4). An accuracy analysis was also performed for the selected biomarkers using receiver operating characteristic (ROC) curves, two of which presented remarkable results. Pyruvate (*m*/*z* = 111) alone had an AUC of 0.974 (Figure 5a), with sensitivity and specificity of 99.2% and 94.6%, respectively (Figure 5b). When pyruvate was combined with 5-hydroxymethyluracil (*m*/*z* = 143) for ROC curve analysis, the graph displayed a better performance, with an AUC of 0.986 (Figure 5c), a sensitivity of 98.3%, and a specificity of 97.2% (Figure 5d).

## 3. Discussion

To assist with the comprehension of GB pathophysiology and identification of biochemical pathways, seven relevant biomarkers with increased levels in the plasma of patients with GB compared with healthy participants were selected. To evaluate their potential for brain tumor screening, a receiver operating characteristic (ROC) curve was constructed for each biomarker and their combinations (Figure 5). Notably, GB plasma samples might comprise pyruvate (*m*/*z* = 111) alone as an important discriminating biomarker (Figure 5a,b) that indicates tumor presence. However, pyruvate is not a GB-exclusive metabolite. Therefore, we evaluated its association with other biomarkers and obtained an interesting ROC curve when pyruvate was combined with 5-hydroxymethyluracil (*m*/*z* 143) (Figure 5c,d). Both metabolites could be useful in the follow-up of patients with GB and could be employed for the early detection of recurrence and establishment of faster diagnostic and therapeutic guidance.

Beyond the appraisal of their biomarker screening potential, the biological roles of these metabolites in human metabolism were evaluated, and interesting correlations were observed as described below.

Among the seven GB chemical markers selected in our analyses, the most frequent metabolite was represented by *m*/*z* = 294 and was identified as the peptide, arginyl-proline (RP). RP is now being first reported as a plasma biomarker in patients with GB. According to some studies, the catabolism of proteins and amino acids is increased in hypoxic GB cells, in accordance with the protein degradation observed in chronic hypoxia [12,13], which might explain the selection of RP as a relevant biomarker. Interestingly, different studies have reported that the dipeptide proline/arginine is associated with neurodegenerative conditions that have been suggested for GB [14]. This aggressive brain tumor is associated with synaptic loss and neurodegeneration, an important component of lethality in GB [15]. Proline and arginine repeats are associated with cell death owing to their toxic effects, which are represented by the binding of chaperones, such as prolyl isomerases, and interference in protein folding [14,16]. Misfolded proteins are defective and dysfunctional, which is concerning, especially if they affect tumor suppressor proteins, which corroborates with cell cycle dysregulation and cancer development [17]. Therefore, the RP dipeptide, proposed as an unedited biomarker (*m*/*z* = 294), might be involved in neurodegeneration associated with GB (Figure 6); however, this hypothesis requires further testing.

In this scenario of molecular alterations, tumoral cells tend to present epigenetic modulation of gene expression through diverse biochemical modifications, such as methylation and demethylation processes. Owing to the aberrant methylation of CpG islands in cancers, DNA-demethylation is known to be involved in epigenetic changes, especially in cancer. One of the selected biomarkers corresponded to *m*/*z* = 143 and was identified as 5-hydroxymethyluracil, which is a byproduct derived from different pathways (Figure 7), including (i) the active DNA-demethylation process through deamination of 5-hydroxymethylcytosine [18] and (ii) the oxidation/hydroxylation of thymine derived from intense oxidative stress [19,20]. These processes can lead to harmful mutagenic alterations, such as base pair exchanges, culminating in tumorigenesis [21]. Some cancers, such as breast and colorectal cancers, are associated with higher levels of anti-5-hydroxymethyluracil antibodies. In fact, in women with breast cancer, higher levels of 5-hydroxymethyluracil have been detected, especially in blood samples [22,23]. Although human cells are susceptive to an imbalance in the redox state, the brain tissue is particularly vulnerable to a hypoxic environment and oxidative stress due to its high demand for oxygen and high expression of superoxide dismutase (SOD), associated with OH- synthesis and its consequences, including metabolic changes and chromosomal instability [24]. SOD overexpression is especially observed in astrocytes [24], the primary source of GB [2], and may serve as one of the reasons for the remarkable aggressiveness of GB and the difficulty associated with treating this tumor compared with other tumors. Together, these data support the hypothesis that 5-hydroxymethyluracil might indicate genomic instability, highlighting a potential plasmatic biomarker for tumor screening for GB. To our knowledge, this is the first study to highlight the presence of 5-hydroxymethyluracil in the plasma of GB patients.

In addition to epigenetic modulation and redox imbalance, energy metabolism is affected in tumorigenesis, which was reinforced by the identification of pyruvate (*m*/*z* = 111) as one of the biomarkers in the present study. Notably, pyruvate is commonly found at increased levels in tumors. This characteristic may be partly explained by the Warburg effect, which refers to metabolic reprogramming in cancer cells that prioritizes alternative energetic routes instead of mitochondrial respiration [25,26]. Mutations associated with enzymes involved in energy metabolism, such as isocitrate dehydrogenase (IDH), are commonly observed in gliomas. IDH1 mutant cells were demonstrated to overexpress pyruvate dehydrogenase kinase 3 (PDK3), a well-known downregulator of the activity of pyruvate dehydrogenase (PDH) [27,28], the enzyme responsible for pyruvate conversion to acetyl CoA. Therefore, glioma cells are prone to increased pyruvate levels, as revealed in the blood samples tested in the present study. 

Although the energy metabolism of cancer cells relies on aerobic glycolysis [29], some studies have shown that gliomagenesis presents glycolysis and fatty acid oxidation (FAO) in a dynamic relationship that is critical for cellular metabolism in the heterogeneous environment of GB cells [30]. McKelvey et al. revealed the upregulation of glycolytic and FAO enzymes in GB tumors. When both energy pathways were targeted in vivo, the survival rate of GB mice was found to increase [31]. Our metabolomic analysis revealed an increased frequency of an FAO metabolite in GB samples, 3-oxodecanoyl-CoA (*m*/*z* = 936), which corroborates the importance of energetic alternatives for the maintenance of the GB environment. Overall, 3-oxodecanoyl-CoA and pyruvate highlight the importance of both glycolysis and FAO dynamics in GB and reinforce the proposal of targeting both pathways to improve conventional therapeutic approaches.

In this scenario, in which metabolic reprogramming occurs via diverse pathways, endocannabinoid biosynthesis may also be affected. Anandamide (AEA), one of the most studied endocannabinoids, is an important neuromodulatory lipid in the central nervous system [32]. AEA is synthesized from N-acylphosphatidylethanolamines (NAPEs) by the NAPE-phospholipase D enzyme (NAPE-PLD), which is less active in GB tissue, leading to decreased levels of anandamide [33]. In accordance with NAPE-PLD-reduced activity, NAPE levels were observed to be higher than those in non-tumor tissues [33], not only for GB but also for different tumors [34], which might indicate a retroactive stimulus for NAPE biosynthesis. In the present study, NAPE was among the most important features of GB plasma samples, represented by *m*/*z* = 982. NAPE prominence indicates that AEA might not be synthesized, and its function in tumors through the cannabinoid receptor, CB1 [35], might be reduced. If not activated, the endogenous levels of cAMP tend to increase and activate protein kinase A, resulting in the activation of downstream signaling pathways, such as mitogen-activated protein kinases (MAPKs), through the cAMP response element-binding protein (CREB), which is associated with cellular proliferation, tumor growth, and invasion [36,37], as shown in Figure 8. As the antitumoral effects of anandamide are frequently reported [38,39], tumoral alterations, either at the gene or biochemical level, may induce the inhibition of AEA biosynthesis and favor GB progression, leading to a worse prognosis.

MAPK signaling is mediated via the serine/threonine Raf kinase, which must interact with membrane-acidic cytoplasmic domains to be activated [40]. The most abundant acidic phospholipid in the brain cortex and exclusively present at the cytoplasmic leaflet plasma membrane is phosphatidylserine (PS) [41]. PS is the critical membrane phospholipid that interacts with Raf kinase, resulting in its activation and consequent ERK phosphorylation, caspase-3 inhibition, and suppression of apoptotic signaling, thereby corroborating tumor survival [42]. In the present study, PS (*m*/*z* = 819) was selected as one of the most important and frequent biomarkers in GB plasma samples, which aligns with the findings of some studies that revealed that higher levels of PS enhance Raf-1 activation [42,43]. This result is consistent with aberrant Ras/MAPK activation and cancer progression [44]; however, this is not the only biochemical pathway affected by increased levels of PS. PI3K/Akt has also been extensively reported to be an important pathway in tumorigenesis [45] and is positively affected by PS. Akt activation depends on its association with PS in the cytoplasmic leaflet, which leads to conformational alterations and exposes the Akt kinase domain to enable phosphorylation by mTORC2, thereby enhancing Akt activation [46,47]. Once activated, Akt phosphorylates diverse cytoplasmic proteins implicated in several downstream tumorigenic events, such as cellular proliferation, control of apoptosis, and invasiveness [48,49] (Figure 8).

Another metabolite known to interfere with tumor progression and the Akt pathway is sphingolipid ceramide, which mediates the downregulation of Akt and elicits mitochondrial apoptotic signaling [50]. However, tumors tend to have reduced levels of ceramide due to intensive metabolism, which enhances Akt signaling and tumorigenesis [51]. Among the GB-abundant features selected in our study, a product of ceramide metabolism, 3-O-sulfogalactosylceramide (a glycosphingolipid; *m*/*z* = 931) was identified. This product is a sulfatide, a class of sulfated glycolipids that is considered the most abundant in the myelin sheath [52]. This compound is composed of a sphingosine C18 carbon chain associated with the fatty acid C24, which is present at increased levels in human glioma cell lines [53,54]. Additionally, the level of 3-O-sulfogalactosylceramide has been found to be increased in the plasma of patients with meningioma [55] and different tumors [56,57,58] and has been associated with metastasis risk [59]. In addition, sulfated glycolipids are known to attach to adhesion molecules, especially P-selectin, one of the main membrane glycoproteins responsible for cell adhesion in endothelial cells, [60] and are overexpressed in GB cells [61]. Therefore, increased levels of 3-O-sulfogalactosylceramide may indicate the occurrence of metastasis associated with a brain tumor and may be a potential biomarker of aggressiveness and invasiveness. 

This study could guide future investigations of altered biochemical pathways in GB. The present biomarkers highlight the different pathways affected. Further, some of these metabolites are described, for the first time, in association with GB, especially in plasma samples. Our findings corroborate the understanding of GB through the establishment of the correlations among biomarkers and the biochemical pathways associated with GB. In addition, our data provide new opportunities to investigate brain tumor pathophysiology and new potential targets for pharmacological interference.

Although we did not evaluate genetic and epigenetic alterations, we attempted to elucidate the intracellular connections of each biomarker with disease pathophysiology and establish well-defined molecular interactions between genes and metabolites. We also used a small number of GB samples, which serves as another limitation of this study. As the incidence of GB remains at approximately 10 cases per 100,000 individuals [2], this result is in accordance with the area where the study was Although it is a preliminary analysis, we recognize more studies with a larger number of samples are necessary to validate our findings, especially if the proposal is to be employed in the biomedical field. In this context, our study highlights new research topics for further investigation, enabling the elucidation of the behavior of GB and an evaluation of the use of these biomarkers in medicine. 

## 4. Materials and Methods

### 4.1. Patient Selection

This study was approved by the Research Ethics Committee of Botucatu Medical School (number 3.491.414; CAAE 16610019.9.0000.5411) and aligned with the guidelines of the Declaration of Helsinki. Blood samples were collected from 50 healthy individuals (Control—CT group) and 24 patients with GB (GB group) at the Neurosurgery Service of the Outpatient Clinic of Botucatu Medical School, UNESP, São Paulo, Brazil. Plasma samples were stored at −72 °C until metabolite extraction and analysis.

### 4.2. Sample Preparation

The protocol by Melo et al. was used to extract the metabolites [62]. Briefly, 20 µL of plasma was poured into 200 µL of tetrahydrofuran, homogenized under vortex, and centrifuged at 3200 rpm for 5 min. Thereafter, the supernatant was collected and poured into 780 µL of methanol. This mixture was then subjected to homogenization and centrifugation using the same parameters mentioned above. A total of 50 µL of the supernatant was solubilized in methanol q.s. 500 µL and homogenized. Finally, 0.1% formic acid was added to the mixture to assist with ionization. 

### 4.3. Mass Spectrometric Analysis

For mass spectrometric analysis, the solution was directly injected into an Electrospray Ionization source and analyzed using an LTQ Mass Spectrometer (ESI-LTQ-XL Discovery, Thermo Scientific, Bremen, Germany) in positive ion mode in the mass range of 100–1400 *m*/*z*. The spectra were analyzed using XCalibur software (v. 2.4; Thermo Scientific, San Jose, CA, USA). Ten analytical spectra were generated for each biological sample, resulting in 240 and 500 spectra for the GB and CT groups, respectively. For spectral data acquisition, the following parameters were employed: capillary temperature, 280 °C; flow rate, 10 μL·min; spray voltage, −15 kV; and sheath gas, 10 arbitrary units. 

### 4.4. Statistical Analysis

The CT and GB datasets were inputted into the MetaboAnalyst 4.0 platform (www.metaboanalyst.ca (accessed on 10 February 2020)) [63] for statistical analysis, which was initially performed using partial least squares discriminant analysis (PLS-DA). As a result, a score plot was generated to observe the separation between the GB and CT groups. The variable importance in projection (VIP) scores established by PLS-DA were used to select the most important metabolites in each group; a VIP score greater than 2.5 was defined as the threshold for the present analysis. Fold-change (FC) and a Student’s ***t***-test analyses were performed to assist in the selection of the most abundant metabolites for each group. FC > 2 was defined as the threshold and used to generate the fold-change graphic using Log^2^FC as a statistical parameter. *p*-value < 0.05 was added to FC analysis and generated a volcano plot. A heat map of the most important and significant biomarkers was generated using Ward’s clustering algorithm and Euclidean distance measurements to illustrate the distribution of the selected metabolites in the samples from each group. To assess the potential of the selected biomarkers as a screening test, accuracy was evaluated using a receiver operating characteristic (ROC) curve and the linear SVM algorithm. A ROC curve was constructed for each biomarker, and the area under the curve (AUC) was evaluated. The statistical parameters of sensitivity and specificity were calculated from the confusion matrix. Only biomarkers with AUC > 0.95 were considered. 

### 4.5. Molecular Identification

The selected *m*/*z* biomarkers for the GB group were identified using *tandem* mass spectrometry (MS/MS) and Mass Frontier software (v. 6.0, Thermo Scientific, San Jose, CA, USA), and analyzed in online metabolomics databases, such as METLIN (http://metlin.scripps.edu), Human Metabolome Database (https://hmdb.ca/ (accessed during the year of 2020)), and LIPID Maps (https://www.lipidmaps.org/ (accessed during the year of 2020)). Altogether, these analyses were the basis for proposing the biological significance of the metabolites and pathways involved in disease.

## Figures and Tables

**Figure 1 ijms-24-08813-f001:**
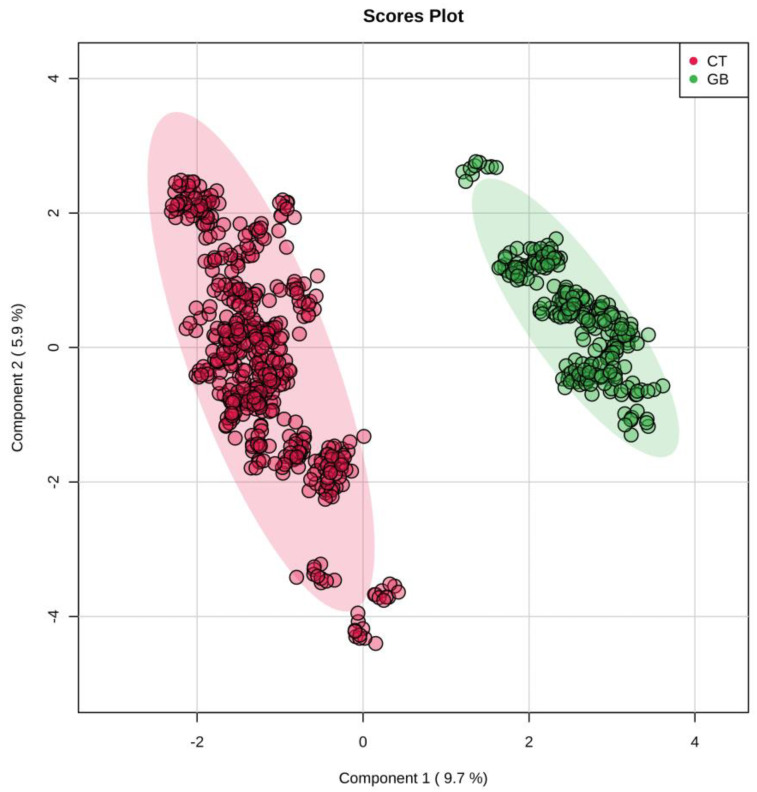
PLS-DA highlighting the ability to discriminate between the GB and CT groups. Score scatter plot based on partial least square-discriminant analysis (PLS-DA) of metabolomics data from serum samples of the control (CT) and glioblastoma (GB) groups, with a clear separation between the groups and clustering of samples from the same group. CT (●) and GB (●) groups.

**Figure 2 ijms-24-08813-f002:**
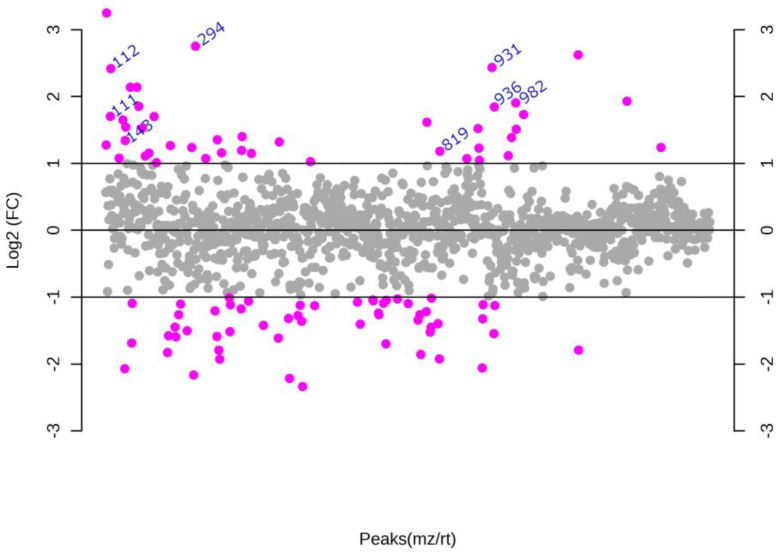
Fold change (FC) representation of the metabolites according to abundance. The diagram was generated based on Log_2_(FC) according to the relative abundance of metabolites (biomarkers) in GB/CT. GB biomarkers were considered when FC > 2.0, i.e., Log_2_(FC) > 1.0. The indicated *m*/*z* values correspond to more abundant biomarkers of GB.

**Figure 3 ijms-24-08813-f003:**
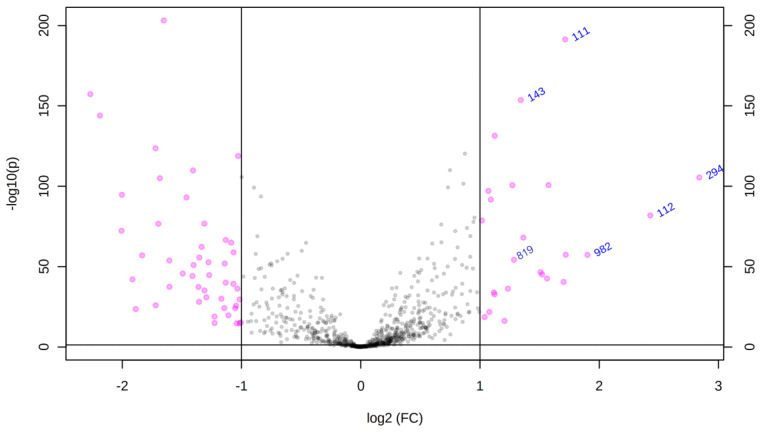
Volcano plot with the most discriminant *m*/*z* values for GB samples. Univariate statistical analysis of the detected metabolites in which their relative abundances (GB/CT) are presented on a volcano plot, where *m*/*z* values with *p* < 0.05 and fold change > 2.0 (−1.0 > Log_2_(FC) > 1.0) were considered significant to discriminate glioblastoma and healthy samples.

**Figure 4 ijms-24-08813-f004:**
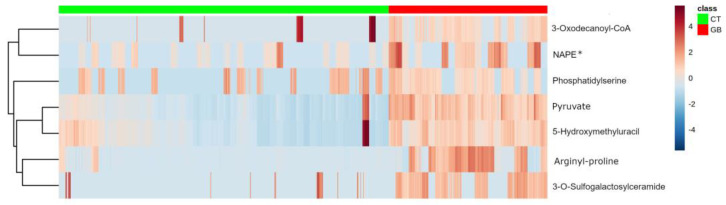
Heatmap depicting the increased intensity of the seven selected GB metabolites in GB patients. Heatmap demonstrating the distribution of metabolites according to the individuals and their groups, represented by superior horizontal green and red bars (Control group—CT: ▬; Glioblastoma group—GB: ▬). On the graphic right-lateral side, each line represents one metabolite, where the color gradation represents the peak intensity from red (more intense) to blue (less intense). * NAPE: N-acylphosphatidylethanolamine.

**Figure 5 ijms-24-08813-f005:**
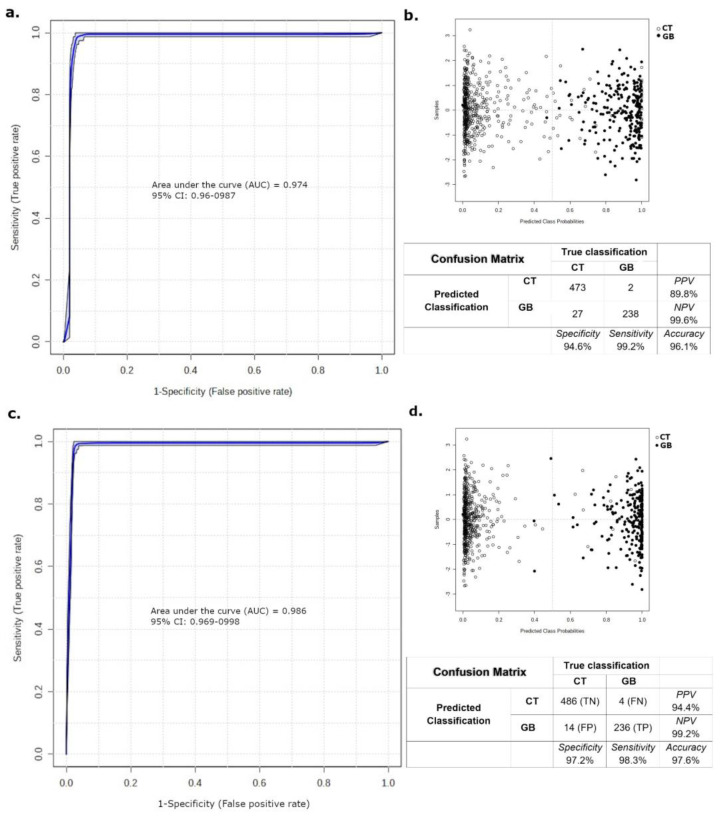
ROC curve presents AUC > 0.95 when performed with Pyruvate alone or combined with 5-hydroxymethyluracil. Analysis of the ROC curve performance with pyruvate (**a**) and its confusion matrix are represented with the statistical parameters (**b**). The same data are presented when pyruvate and 5-hydroxymethyluracil were evaluated together, whose ROC curve (**c**), confusion matrix, and statistical parameters are shown in (**d**).

**Figure 6 ijms-24-08813-f006:**
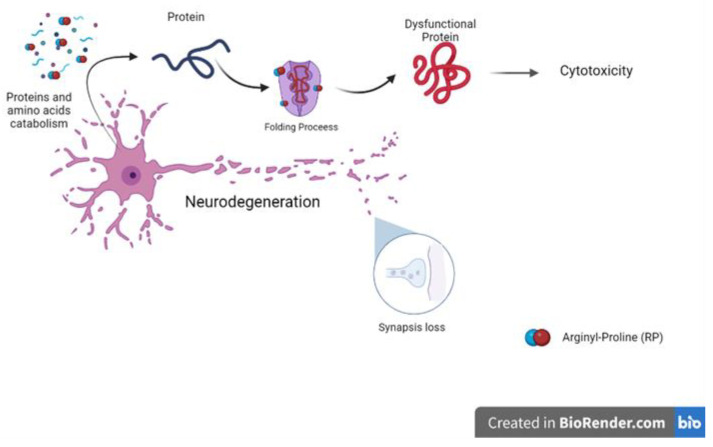
Scheme illustrating the dysfunctional effects of the dipeptide, Proline/Arginine, on glial cells.

**Figure 7 ijms-24-08813-f007:**
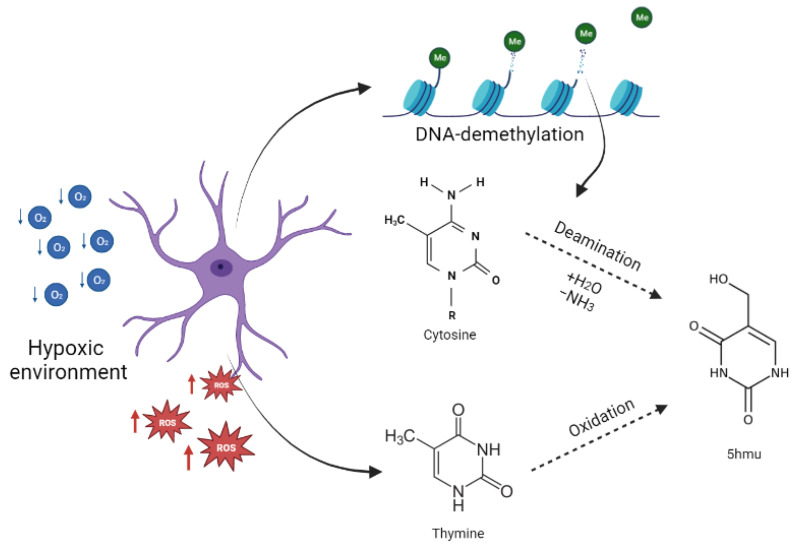
Scheme illustrating the origin of 5-hydroxymethyluracil (5hmu) in a hypoxic environment.

**Figure 8 ijms-24-08813-f008:**
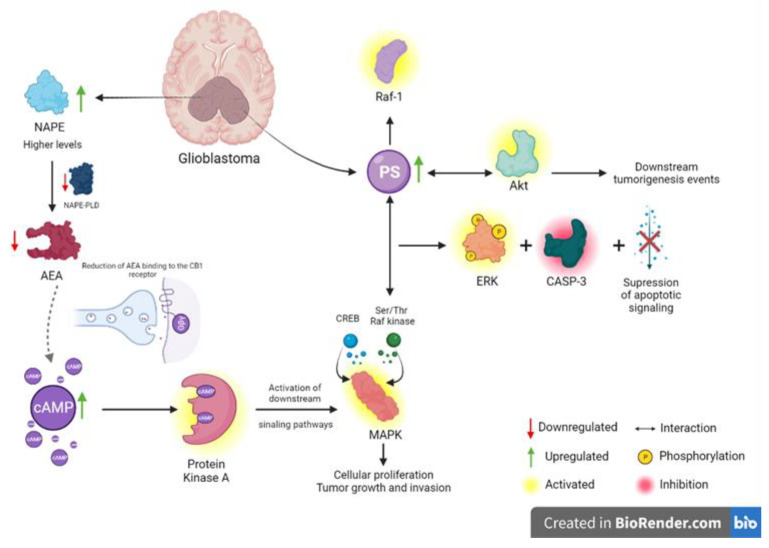
Scheme illustrating the dysfunctional participation of phosphatidylserine (PS) and N-acylphosphatidylethanolamines (NAPEs) in glial metabolic reprogramming.

**Table 1 ijms-24-08813-t001:** Identity data of the proposed chemical markers for glioblastoma according to the *m*/*z* features selected based on the statistical parameters of VIP score > 2.5 (PLS-DA) and fold change > 2.0.

*m*/*z*	ID	Metabolite	Molecular Formula	Adduct	MS/MS ^c^	Log_2_FC ^d^
111	MID 117 ^a^	Pyruvate	C_7_H_12_O_2_	[M+ Na]^+^	69, 55, 93, 83	2.4179
143	MID 5456 ^a^	5-Hydroxymethyluracil	C_5_H_6_N_2_O_3_	[M + H]^+^	116, 111, 117, 97, 125	1.3418
294	MID 85632 ^a^	Arginyl-Proline	C_11_H_21_N_5_O_3_	[M + Na]^+^	268, 254, 250, 236, 266	2.7526
819	MID 78327 ^a^	Phosphatidylserine (38:9) ^b^	C_44_H_68_NO_10_P	[M + NH_4_]^+^	184, 636, 760, 147, 113	1.1817
931	MID 5096 ^a^	3-O-Sulfogalactosylceramide (42:1) ^b^	C_48_H_93_NO_11_S	[M + K]^+^	109, 112, 121, 135, 184	2.4353
936	MID 58189 ^a^	3-Oxodecanoyl-CoA	C_31_H_52_N_7_O_18_P_3_S	[M + H]^+^	522, 184, 113	1.8450
982	MID 76593 ^a^	NAPE (N-acylphosphatidylethanolamine)	C_57_H_108_NO_9_P	[M + H]^+^	644, 360, 113, 184	1.9059

^a^ METLIN Representative ID; ^b^ carbon number: double bond; ^c^ fragmentation profile of tandem mass spectrometry; ^d^ fold change; molecules with the same *m*/*z* and a similar fragmentation profile.

## Data Availability

The data presented in this study are available on request from the corresponding author. The data are not publicly available due to ethical and privacy restrictions.

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
