# Peer review of "Metabolomics Approach Reveals Important Glioblastoma Plasma Biomarkers for Tumor Biology"

_ijms, 2023, doi:10.3390/ijms24108813_

Round 1
Reviewer 1 Report
The manuscript is interesting and the presented data, even predictable, are important in the field and may be good source for further citations. Furthermore, limitations should be expanded on to address further limitations of this study design and more specific directions for future research provided.
I would consider the paper for publication after the authors have addressed the major questions.
Introduction
In the introduction, the authors should describe primary and secondary goals.
Another limitation of this study is the small number of patients recruited, which may have limited the power of the statistical analysis.
There should be more comparisons of your research with that of other authors in the discussion.
Author Response
We appreciate the review of our manuscript and are very pleased with the level of attention and detail of the review. Considering reviewer’s suggestion, we have rewritten the goals as observed in the revised manuscript. About the sample number, this work was based on previous studies with the same theme and published in journals considered relevant by the scientific community. Besides, glioblastoma cases are relatively rare, in which the incidence varies between 3.26 cases per 100,000 person-years (Ostrom et al., 2022).
Thank you for the suggestion of including “more comparisons of our research with that of other authors in the discussion”, and we take the opportunity to clarify that we exhausted all the possibilities of comparisons with available data on the scientific literature about the theme.
Reference:
Ostrom QT, Price M, Neff C, Cioffi G, Waite KA, Kruchko C, Barnholtz-Sloan JS. CBTRUS Statistical Report: Primary Brain and Other Central Nervous System Tumors Diagnosed in the United States in 2015-2019. Neuro Oncol. 2022 Oct 5;24(Suppl 5):v1-v95. doi: 10.1093/neuonc/noac202. PMID: 36196752; PMCID: PMC9533228.
Reviewer 2 Report
The research work conducted is very interesting can prove to be off great interest as the authors try to elucidate various biomarkers in glioblastoma. It is also very interesting to read how omics was used in this case to isolate metabolites to elucidate various pathways. My comments are the following;
1. The title is too vague and can be a bit more specific and sounds redundant
2. In the introduction, there are so many single citations, alot of major statements were made with single to no citation, please include more.
3. Would it be possible to include a volcano plot to get a better understanding of you mass spec data along with a venn diagram based on your epidemiological characteristics.
4. The authors have only done mass spec, but the discussion elaborates on alot of possibilities with no backing from actual data. I would strongly advise authors to discuss their data and speculations relevant to their findings. It would benefit the authors if they have a specific signaling pathway in mind, they can perform a western blot to confirm.

Author Response
We are grateful for the recognition of our job. The title of the manuscript was modified according to the reviewer’s suggestion. We appreciate the suggestion and have included the requested graphic in the results. We also appreciate the suggestions and recognize the importance of validating these results, and we intend to carry it out further by expanding it through in vitro laboratorial assays. For that, we purpose to evaluate cells cultures of glioblastoma cells and healthy glial cells using silencing or overexpression techniques of key biological pathways involved with the metabolites pointed in the present study. Besides, we also intend to perform quantitative laboratory assays, such as Real Time PCR. Nonetheless, these analyses require considerable time, therefore, we would like to publish the present study as it is for now, sharing these results with other researchers and encouraging further studies with samples from different populations.
Reviewer 3 Report
The development of noninvasive, highly accurate diagnostics is a rapidly growing field, and this paper makes an important contribution. The data are novel and informative.
1) The Discussion, however, is not well organized. About the pathways, the Data could be interpreted in a number of ways. Presenting the hypothesis or model should be more restrained and rewritten into a more compact description.
2) Shouldn't the metabolic effects of the patient's medications be considered?
3) If you have information on the time of sample collection, you should consider the effects of diurnal variations in metabolism.
Please revise the entire English sentence from a grammatical point of view: there are several places where "what" is used as the relational pronoun "which"; the word "inedited" in the title may be used to mean novel, but it is a word not found in common dictionaries (OLED, COUBILD, etc.).
Author Response
We appreciate the review and are very happy with the level of attention to the manuscript. We also take the opportunity to clarify that the identified metabolites do not participate of the same biochemical pathways, therefore we choose the most important pathways associated with them intending to comprehend their role in tumor biology.
Considering the observation “of the metabolic effects of the patient's medications”, the chosen database presents bioinformatics resources to eliminate the main metabolites originated from medications.
Considering the time of sample collection, the patients were submitted to surgical procedures and sample collection preferably in the morning and, in rare exceptions, in the early afternoon.
The authors thank the suggestion of improving the English. Intending to meet the suggestion and refine the manuscript, we have sent it to an English editing and proofreading service – EDITAGE for English revision. The editing service certificate is attached, and we believe the English writing is now adequate.